# Active Learning in CNNs via Expected Improvement Maximization

## Abstract

Deep learning models such as Convolutional Neural Networks (CNNs) have demonstrated high levels of effectiveness in a variety of domains, including computer vision and more recently, computational biology. However, training effective models often requires assembling and/or labeling large datasets, which may be prohibitively time-consuming or costly. Pool-based active learning techniques have the potential to mitigate these issues, leveraging models trained on limited data to selectively query unlabeled data points from a pool in an attempt to expedite the learning process. Here we present "Dropout-based Expected IMprOvementS" (DEIMOS), a flexible and computationally-efficient approach to active learning that queries points that are expected to maximize the model's improvement across a representative sample of points. The proposed framework enables us to maintain a prediction covariance matrix capturing model uncertainty, and to dynamically update this matrix in order to generate diverse batches of points in the batch-mode setting. Our active learning results demonstrate that DEIMOS outperforms several existing baselines across multiple regression and classification tasks taken from computer vision and genomics.

## 1 Introduction

Deep learning models (LeCun et al., 2015) have achieved remarkable performance on many challenging prediction tasks, with applications spanning computer vision (Voulodimos et al., 2018), computational biology (Angermueller et al., 2016), and natural language processing (Socher et al., 2012). However, training effective deep learning models often requires a large dataset, and assembling such a dataset may be difficult given limited resources and time.

Active learning (AL) addresses this issue by providing a framework in which training can begin with a small initial dataset and, based on an objective function known as an acquisition function, choosing what data would be the most useful to have labelled (Settles, 2009). AL has successfully streamlined and economized data collection across many disciplines (Warmuth et al., 2003; Tong & Koller, 2001; Danziger et al., 2009; Tuia et al., 2009; Hoi et al., 2006; Thompson et al., 1999). In particular, pool-based AL selects points from a given set of unlabeled pool points for labelling by an external oracle (e.g. a human expert or biological experiment). The resulting labeled points are then added to the training set, and can be leveraged to improve the model and potentially query additional pool points (Settles, 2011).

Until recently, few AL approaches have been formulated for deep neural networks such as CNNs due to their lack of efficient methods for computing predictive uncertainty. Most acquisition functions used in AL require reliable estimates of model uncertainty in order to make informed decisions about which data labels to request. However, recent developments have led to the possibility of computationally tractable predictive uncertainty estimation in deep neural networks. In particular, a framework for deep learning models has been developed viewing dropout (Srivastava et al., 2014) as an approximation to Bayesian variational inference that enables efficient estimation of predictive uncertainty (Gal & Ghahramani, 2016).

Our approach, which we call "Dropout-based Expected IMprOvementS" (DEIMOS), builds upon prior work aiming to make statistically optimal AL queries by selecting those points that minimize expected test error (Cohn et al., 1996; Gorodetsky & Marzouk, 2016; Binois et al., 2019; Roy & McCallum, 2001). We extend such approaches to CNNs through a flexible and computationally

efficient algorithm that is primarily motivated by the regression setting, for which relatively few AL methods have been proposed, and extends to classification.

Many AL approaches query the *single* point in the pool that optimizes a certain acquisition function. However, querying points one at a time necessitates model retraining after every acquisition, which can be computationally-expensive, and can lead to time-consuming data collection (Chen & Krause, 2013). Simply greedily selecting a certain number of points with the best acquisition function values typically reduces performance due to querying similar points (Sener & Savarese, 2018).

Here we leverage uncertainty estimates provided by dropout in CNNs to create a dynamic representation of predictive uncertainty across a large, representative sample of points. Importantly, we consider the full joint covariance rather than just point-wise variances. DEIMOS acquires the point that maximizes the expected reduction in predictive uncertainty across all points, which we show is equivalent to maximizing the expected improvement (EI). DEIMOS extends to batch-mode AL, where batches are assembled sequentially by dynamically updating a representation of predictive uncertainty such that each queried point is expected to result in a significant, non-redundant reduction in predictive uncertainty. We evaluate DEIMOS and find strong performance compared to existing benchmarks in several AL experiments spanning handwritten digit recognition, alternative splicing prediction, and face age prediction.

## 2 RELATED WORK

AL is often formulated using information theory (MacKay, 1992b). Such approaches include querying the maximally informative batch of points as measured using Fisher information in logistic regression (Hoi et al., 2006), and Bayesian AL by Disagreement (BALD) (Houlsby et al., 2011), which acquires the point that maximizes the mutual information between the unknown output and the model parameters.

Many AL algorithms have been developed based on uncertainty sampling, where the model queries points about which it is most uncertain (Lewis & Catlett, 1994; Juszczak & Duin, 2003). AL via uncertainty sampling has been applied to SVMs using margin-based uncertainty measures (Joshi et al., 2009). AL has also been cast as an uncertainty sampling problem with explicit diversity maximization (Yang et al., 2015) to avoid querying correlated points.

EI has been used as an acquisition function in Bayesian optimization for hyperparameter tuning (Eggensperger et al., 2013). Other AL objectives similar to ours have been explored in (Cohn et al., 1996; Gorodetsky & Marzouk, 2016; Binois et al., 2019; Roy & McCallum, 2001), making statistically optimal queries that minimize expected prediction error, which often reduces to querying the point that is expected to minimize the learner's variance integrated over possible inputs. DEIMOS extends EI and integrated variance approaches, which have traditionally been applied to Gaussian Processes, mixtures of Gaussians, and locally weighted regression, to deep neural networks.

Until recently, few AL approaches have proven effective in deep learning models such as CNNs, largely due to difficulties in uncertainty estimation. Although Bayesian frameworks for neural networks (MacKay, 1992a; Neal, 1995) have been widely studied, these methods have not seen widespread adoption due to increased computational cost. However, theoretical advances have shown that dropout, a common regularization technique, can be viewed as performing approximate variational inference and enables estimation of model and predictive uncertainty (Gal & Ghahramani, 2016). Simple dropout-based AL objectives in CNNs have shown promising results in computer vision classification applications (Gal et al., 2017).

Several new algorithms show promising results for batch-mode AL on complex datasets. Batchbald (Kirsch et al., 2019) extends BALD to batch-mode while avoiding redundancy by greedily constructing a query batch that maximizes the mutual information between the joint distribution over the unknown outputs and the model parameters. Batch-mode AL in CNN classification has also been formulated as a core-set selection problem (Sener & Savarese, 2018) with data points represented (embedded) using the activations of the model's penultimate fully-connected layer. The queried batch of points then corresponds to the centers optimizing a robust (i.e. outlier-tolerant) k-Center objective for these embeddings.

In spite of these recent advances, there are no frameworks for batch-mode AL in CNNs that model the full joint (rather than point-wise) uncertainty and do not require a vector space embedding of the data. Additionally, few AL approaches for CNNs have been assessed (or even proposed) for regression as compared to classification.

## 3 ACTIVE LEARNING VIA EXPECTED IMPROVEMENT (EI) MAXIMIZATION

### 3.1 MOTIVATION

Our pool-based AL via EI maximization framework, DEIMOS, aims to maximally reduce expected (squared) prediction error on a large, representative sample of points by querying as few points from the pool as possible. Let $D_{\text{samp}} = (X_{\text{samp}}, Y_{\text{samp}})$ be an sufficiently large random sample from the training and pool points that it can be assumed representative of the dataset as a whole. Let $D_{\text{pool}} = (X_{\text{pool}}, Y_{\text{pool}})$ denote the available pool of unlabelled data, $(x_{\text{new}}, y_{\text{new}}) \in D_{\text{pool}}$ a candidate point for acquisition, $\theta$ the model parameters with approximate posterior $q(\theta)$, and $\hat{y}_i(\theta)$ the model prediction for an input $x_i$. Note that $Y_{\text{pool}}$ is not known and $Y_{\text{samp}}$ is only partially known in AL as the pool consists of unlabeled points. For brevity, conditioning on model input is omitted in subsequent equations (e.g. $\mathbb{E}_q[\hat{y}_i(\theta) \mid x_{\text{new}}, x_i]$ will be written as $\mathbb{E}_q[\hat{y}_i(\theta) \mid x_{\text{new}}]$).

We seek to maximize EI by acquiring the pool point that minimizes expected prediction error on $D_{\text{samp}}$. The expected prediction error on $D_{\text{samp}}$ conditioned on some $x_{\text{new}} \in X_{\text{pool}}$ is,

$$\frac{1}{|Y_{\text{samp}}|} \sum_{(x_i, y_i) \in D_{\text{samp}}} \mathbb{E}_{q,n}[(y_i - \hat{y}_i(\theta))^2 \mid x_{\text{new}}] =$$

$$\frac{1}{|Y_{\text{samp}}|} \sum_{(x_i, y_i) \in D_{\text{samp}}} \mathbb{E}_q[(\hat{y}_i(\theta) - \mathbb{E}[\hat{y}_i(\theta)])^2 \mid x_{\text{new}}] + (\mathbb{E}_q[\hat{y}_i(\theta) \mid x_{\text{new}}] - \mathbb{E}_n[y_i])^2 + \mathbb{E}_n[(y_i - \mathbb{E}_n[y_i])^2]$$

(1)

where $\mathbb{E}_q$ and $\mathbb{E}_n$ denote expectation over $q(\theta)$ and observation noise, respectively, and $\mathbb{E}_{q,n}$ denotes joint expectation over $q(\theta)$ and observation noise. Here we see the terms contributing to the model's expected prediction error on $D_{samp}$ are (from left to right): 1) the trace of the predictive variance matrix, 2) the sum of the predictive biases squared, and 3) the sum of the noise variances across all observations (a constant in $x_{\text{new}}$) (Cohn et al., 1996). For a purely supervised model, expected predictions remain the same unless the training set of input-output pairs is modified. Therefore, expected model predictions are unchanged conditioned on any unlabeled point $x_{\text{new}}$: $\mathbb{E}_q[\hat{y}_i(\theta) \mid x_{\text{new}}] = \mathbb{E}_q[\hat{y}_i(\theta)]$, and the predictive bias squared for any point $(x_i, y_i)$ stays the same conditioned on $x_{\text{new}}$: $(\mathbb{E}_q[\hat{y}_i(\theta) \mid x_{\text{new}}] - \mathbb{E}_n[y_i])^2 = (\mathbb{E}_q[\hat{y}_i(\theta)] - \mathbb{E}_n[y_i])^2$. Substituting accordingly in equation 1, the pool point that would minimize expected prediction error on $D_{\text{samp}}$ if queried is:

$$x^* = \arg\min_{x_{\text{new}} \in X_{\text{pool}}} \frac{1}{|Y_{\text{samp}}|} \text{tr}(\text{Var}_q(\hat{Y}_{\text{samp}}(\theta) \mid x_{\text{new}}))$$

(2)

where $\hat{Y}_{\text{samp}}(\theta)$ denotes the model predictions for input $X_{\text{samp}}$ and fixed parameters $\theta$ and $\text{Var}_q$ denotes variance over the approximate posterior. Therefore, the $x_{\text{new}}$ that minimizes expected prediction error upon being queried is that expected to minimize average prediction variance (specifically the trace of the predictive covariance matrix). Under our assumptions, knowing $x_{\text{new}}$ is sufficient to calculate the predictive variance contribution to expected prediction error conditioned on $(x_{\text{new}}, y_{\text{new}})$, even when $y_{\text{new}}$ is unknown. Importantly, it follows that, even though $Y_{\text{pool}}$ is not known in AL, DEIMOS is still able to identify and query the pool point $x_{\text{new}} \in X_{\text{pool}}$ that minimizes expected prediction error across $D_{\text{samp}}$ after $(x_{\text{new}}, y_{\text{new}})$ is incorporated into the training set.

### 3.2 MC DROPOUT VARIANCE ESTIMATION

The proposed approach can be implemented with any uncertainty estimation method that enables calculation of the variance and covariance of model predictions. In our experiments, we use MC dropout (Gal & Ghahramani, 2016) to obtain estimates of model uncertainty in CNNs. Models are trained with dropout preceding all fully-connected layers and dropout is used at test time to generate $T$ approximate samples from the posterior predictive distribution. The predictive mean is obtained as the average of these samples.

In order to estimate the predictive covariance matrix for all sample points, $J$ dropout masks are randomly generated for all dropout layers in the neural network. Crucially, the same $J$ dropout masks are used to make test-time predictions *across all sample points* to enable estimating correlation between them. From the dropout sample covariance matrix $\text{Var}_q(\hat{Y}_{\text{samp,dropout}}(\theta))$ capturing the sample variances and covariances of the $J$ dropout predictions for each sample point, one can estimate the prediction covariance matrix (Gal & Ghahramani, 2016),

$$\text{Var}_q(\hat{Y}_{\text{samp}}(\theta)) = \tau^{-1}I + \text{Var}_q(\hat{Y}_{\text{samp,dropout}}(\theta)). \tag{3}$$

Here $\tau = \frac{(1-p)l^2}{2N\lambda}$ represents the model precision in regression tasks, where $p$ is the dropout probability, $l$ is a prior length-scale parameter, $N$ is the number of training points, and $\lambda$ represents the weight decay (Gal & Ghahramani, 2016). In the classification setting, $\tau^{-1} = 0$.

## 3.3 DEIMOS ACTIVE LEARNING IN CNN REGRESSION

DEIMOS queries the pool point $x_{\text{new}}$ that minimizes the expected prediction variance across $D_{\text{samp}}$ upon being queried and, by equation 2, minimizes the expected prediction error on $Y_{\text{samp}}$ upon querying $y_{\text{new}}$. Assuming that all $\hat{y}_i \in \hat{Y}_{\text{samp}}$ are jointly Gaussian, and considering candidate point for acquisition $x_{\text{new}} \in X_{\text{pool}}$:

$$\text{Var}_q(\hat{Y}_{\text{samp}}(\theta) \mid x_{\text{new}}) = \text{Var}_q(\hat{Y}_{\text{samp}}(\theta)) - \text{Cov}_q(\hat{Y}_{\text{samp}}(\theta), \hat{y}_{\text{new}}(\theta))\text{Var}_q(\hat{y}_{\text{new}}(\theta))^{-1}\text{Cov}_q(\hat{Y}_{\text{samp}}(\theta), \hat{y}_{\text{new}}(\theta))^T \tag{4}$$

where $\text{Cov}_q(\hat{Y}_{\text{samp}}(\theta), \hat{y}_{\text{new}}(\theta))$ is a $|Y_{\text{samp}}| \times 1$ column vector. Therefore, in order to maximize the expected model improvement and minimize the expected prediction error on $Y_{\text{samp}}$, DEIMOS queries the point:

$$x^* = \underset{x_{\text{new}} \in X_{\text{pool}}}{\arg\min} \text{tr}(\text{Var}_q(\hat{Y}_{\text{samp}}(\theta) \mid x_{\text{new}}))$$

$$= \underset{x_{\text{new}} \in X_{\text{pool}}}{\arg\max} \text{tr}(\text{Cov}_q(\hat{Y}_{\text{samp}}(\theta), \hat{y}_{\text{new}}(\theta))\text{Var}_q(\hat{y}_{\text{new}}(\theta))^{-1}\text{Cov}_q(\hat{Y}_{\text{samp}}(\theta), \hat{y}_{\text{new}}(\theta))^T) \tag{5}$$

The EI in regression is defined to be the quantity maximized in equation 5: the total reduction in predictive variance across sample points upon querying a given pool point.

## 3.4 DEIMOS ACTIVE LEARNING IN CNN CLASSIFICATION

In the classification setting, test-time dropout remains applicable as a means of drawing from the approximate model posterior. Here the DEIMOS approach for regression is extended to maximizing the expected reduction in uncertainty in predicted class probabilities in classification. Assuming all predicted class probabilities across all sample points, denoted by $\hat{p}_{\text{samp}}$, are jointly Gaussian:

$$\text{Var}_q(\hat{p}_{\text{samp}}(\theta)|x_{\text{new}}) = \text{Var}_q(\hat{p}_{\text{samp}}(\theta))$$
$$- \text{Cov}_q(\hat{p}_{\text{samp}}(\theta), \hat{p}_{\text{new}}(\theta))(\text{Var}_q(\hat{p}_{\text{new}}(\theta)) + \tau_s^{-1}I)^{-1}\text{Cov}_q(\hat{p}_{\text{samp}}(\theta), \hat{p}_{\text{new}}(\theta))^T \tag{6}$$

where $c$ is the number of classes, $\text{Var}_q(\hat{p}_{\text{samp}}(\theta))$ is a $c|Y_{\text{samp}}| \times c|Y_{\text{samp}}|$ matrix, $\text{Cov}_q(\hat{p}_{\text{samp}}(\theta), \hat{p}_{\text{new}}(\theta))$ is a $c|Y_{\text{samp}}| \times c$ matrix, $\text{Var}_q(\hat{p}_{\text{new}}(\theta))$ is a $c \times c$ matrix, and $\tau_s^{-1} \ll 1$ is a smoothing parameter ensuring the invertibility of $\text{Var}_q(\hat{p}_{\text{new}}(\theta))$. We define EI in classification as the total reduction in variance of predicted class probabilities across all sample points that is expected upon querying a point, which is given in equation 6 by: $\text{tr}(\text{Cov}_q(\hat{p}_{\text{samp}}(\theta), \hat{p}_{\text{new}}(\theta))(\text{Var}_q(\hat{p}_{\text{new}}(\theta)) + \tau_s^{-1}I)^{-1}\text{Cov}_q(\hat{p}_{\text{samp}}(\theta), \hat{p}_{\text{new}}(\theta))^T)$. All quantities in equation 6 are estimated using MC dropout equation 3. DEIMOS maximizes the EI by querying the point $x_{\text{new}} \in X_{\text{pool}}$ that minimizes $\text{tr}(\text{Var}_q(\hat{p}_{\text{samp}}(\theta)|x_{\text{new}}))$, where $\text{Var}_q(\hat{p}_{\text{samp}}(\theta)|x_{\text{new}})$ is given by equation 6.

Treating the predicted class probabilities $\hat{p}_{\text{samp}}$ as jointly Gaussian may seem a poor approximation as probabilities are bounded. However, we find that DEIMOS performs better in the bounded probability space than in the unbounded logit space, possibly because in the logit space unimportant differences between large positive (or negative) predictions that correspond to small differences in predicted probabilities are given undue importance.

Applying the joint Gaussian assumption to the class probabilities for all classes across all sample points provides an expression for the probability covariance matrix conditioned on each unlabeled candidate point. DEIMOS acquires the point that maximizes the expected reduction in predictive uncertainty across all classes and all points.

# 4   DEIMOS BATCH-MODE ACTIVE LEARNING

---

**Algorithm 1** DEIMOS batch-mode active learning in regression

---

**Input:** $X_{\text{samp}}$, $X_{\text{train}}$, fixed-mask dropout predictions $\hat{Y}_{\text{samp,dropout}}$, batch size $b$, model precision $\tau$

   $S = |X_{\text{samp}}|$, $X_{\text{batch}} = \varnothing$

   $V = [V_1 \quad \ldots \quad V_S] = \text{Var}_q(\hat{Y}_{\text{samp}}(\theta)) = \tau^{-1}I + \text{Var}_q(\hat{Y}_{\text{samp,dropout}}(\theta))$

   **for** $i \in \{1, \ldots, b\}$ **do**

      $X_{\text{cand}} = X_{\text{samp}} \setminus (X_{\text{train}} \cup X_{\text{batch}})$

      $x_n = \arg\max_{x_i \in X_{\text{cand}}} \text{tr}\left( V_i(V_{ii}^{-1})V_i^T \right)$

      $V = V - V_n(V_{nn}^{-1})V_n^T$

      $X_{\text{batch}} = X_{\text{batch}} \cup x_n$

   **end for**

   **return** $X_{\text{batch}}$

---

The joint Gaussian assumption, applied to real-valued outputs in regression or class probabilities in classification, enables estimation of the predictive covariance across the representative sample of points conditioned on unlabeled points. In the batch-mode setting, DEIMOS assembles each batch sequentially. After a point is added to the batch, having maximized the DEIMOS acquisition function, the predictive covariance matrix is updated conditioned on the unlabeled point; the updated covariance matrix, in turn, is used to calculate the acquisition function values, determine the subsequent point in the batch, and again update the predictive covariance matrix (Algorithm 1). The general procedure of Algorithm 1 applies in classification and regression, but in classification $\tau^{-1} = 0$ and $\tau_s^{-1}$ is introduced, $V$ is a $cS \times cS$ matrix, $V_i$ is a $cS \times c$ matrix, and $V_{ii}$ is a $c \times c$ matrix. Candidate points for acquisition are limited to the unlabeled points in $X_{\text{samp}}$ as opposed to all of $X_{\text{pool}}$ in Algorithm 1 and in our AL experiments in the interest of computational efficiency.

The DEIMOS approach to batch-mode AL provides a computationally efficient mechanism for batch selection. Assembling batches sequentially and updating the covariance matrix after each point is added ensures that all acquired points result in relatively high reductions in predictive uncertainty and that no two points in the queried batch are redundant in their reduction of predictive uncertainty across the sample points. For some $\text{Var}_q(\hat{Y}_{\text{samp}}(\theta))$, there can exist batches of $b$ points that result in a greater variance reduction than the $b$ points queried by the proposed greedy algorithm. However, simulation experiments for small batch sizes show that the variance reduction resulting from our greedy algorithm is typically very close to the optimal variance reduction (Appendix A).

# 5   EXPERIMENTS

First, we visualize MC dropout uncertainty estimates and the proposed acquisition function for synthetic data in a one-dimensional input space. Next, we evaluate DEIMOS empirically via experimental comparison with other acquisition functions and approaches.

We tested DEIMOS in CNNs across regression and classification tasks, and in single-point acquisition and batch-mode AL settings[1]. Specifically, DEIMOS is compared to existing methods in alternative splicing prediction (Rosenberg et al., 2015) and face age prediction (susanqq, 2017) regression tasks. Classification experiments were run on MNIST (LeCun et al., 1998) binary classification of handwritten 7s and 9s, and on classification of all 10 digits. All AL results shown are averaged over three experiments, each beginning with a new, randomly-selected training set. AL

---

[1]AL experiment details are detailed in Appendix D and hyperparameter tuning details are discussed in Appendix E. Code for all experiments is included in the supplementary material.

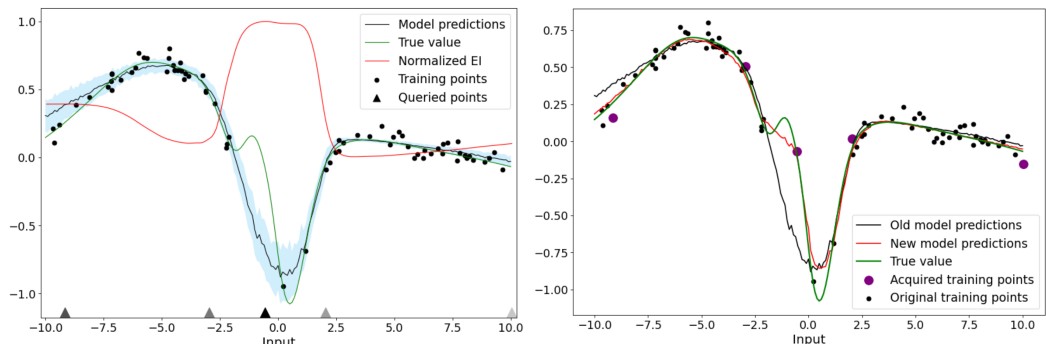

Figure 1: **Left**: DEIMOS acquisition function and batch query for simulated 1D data. Model predictions $\pm 1$ standard deviation are shaded light blue. Points selected to be queried in a batch of size 5 are indicated on the x-axis (batch construction order: darker to lighter markers). **Right**: Model predictions after acquisition. The queried points from 1(a) are labeled and added to the training set and a new model is retrained from scratch. A substantially improved fit is obtained.

Table 1: Number of single-point acquisitions required in order to achieve the 5' splicing test MSE listed in the left column for each acquisition function.

| Test MSE | DEIMOS | Max. variance acquisition | Random acquisition |
|----------|--------|---------------------------|--------------------|
| 0.11 | 69 | 92 | 80 |
| 0.10 | 109 | 318 | 180 |
| 0.095 | 174 | 492 | 306 |
| 0.09 | 369 | 708 | 387 |

performance is compared across algorithms using a linear mixed model to account for dependence across acquistion iterations (Appendix C).

## 5.1 Visualizing DEIMOS acquisition in 1D neural network regression

In order to visualize the DEIMOS acquisition function we simulated a 1D neural network regression task. Synthetic data with real-valued input and output is produced by a random dense neural network and then used to train a dense neural network (Appendix D).

The DEIMOS acquisition function exhibits desirable properties for an AL objective. It is high in $[-2.5, 2.5]$, where data is comparatively sparse, and $[-10, -7]$, where the current model does not fully capture the underlying trend (Fig 1 left). In the $[2.5, 10]$ input region where the model predictions are fairly accurate, EI is relatively low. DEIMOS batch-mode AL queries a diverse set of points, concentrating on the $[-3, 3]$ input region but also querying two points in the surrounding regions (Fig 1 left). In this example, DEIMOS batch-mode AL successfully improves model performance (Fig 1 right), querying points that are expected to reduce predictive uncertainty but are not redundant. Indeed, upon labeling of the requested points, model fit significantly improves in the $[-10, -7]$ and $[-2.5, 2.5]$ input regions.

## 5.2 Active learning experiments: CNN regression

We evaluated DEIMOS on prediction of splice donor site in 5' alternative splicing (Rosenberg et al., 2015) and face age prediction for images in the UTKFace dataset (susanqq, 2017). In each experiment, all acquisition functions start with the same initial training set, and, in each AL iteration, acquire a specified number of points ("batch size") and the models are retrained from scratch. The performance of DEIMOS is compared to that of random acquisition, and to acquiring the candidate pool point(s) with maximum MC dropout variance.

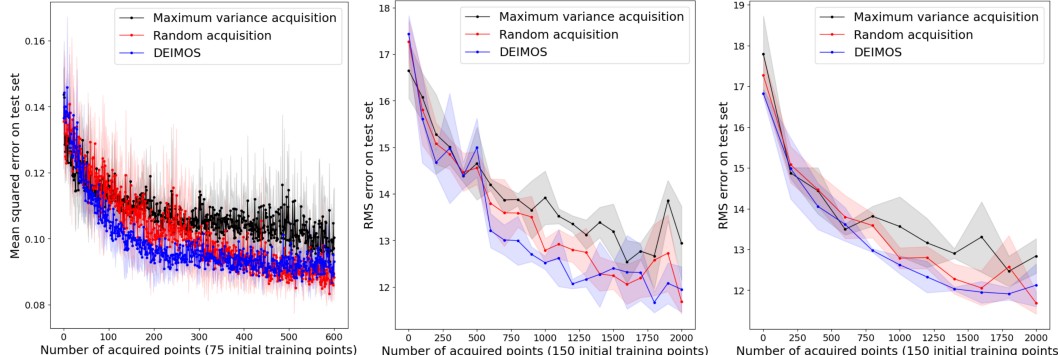

Figure 2: Test performance across AL iterations with $\pm 1$ standard deviation of average performance shaded. **Left**: 5' splicing (batch size 1) MSE; DEIMOS outperforms benchmarks ($p = 2.1 \times 10^{-278}$ for maximum variance acquisition and $p = 6.9 \times 10^{-34}$ for random acquisition) (Appendix C). **Center**: Face age prediction RMSE (batch size 100); DEIMOS outperforms benchmarks ($p < 8 \times 10^{-3}$) (Appendix C). **Right**: Face age prediction RMSE (batch size 200); DEIMOS outperforms benchmarks ($p < 9 \times 10^{-3}$, Appendix C).

**Alternative RNA splicing prediction.** CNN regression models are trained to predict the relative usage of one of at two alternative splice donor sites. One-hot encoded RNA sequences of 101 nucleotides for each set of splicing measurements are used as the model input.

Results are illustrated in Fig 2 (left). DEIMOS outperforms both benchmarks over the first several hundred iterations but random acquisition catches up to DEIMOS near the end of the experiment. DEIMOS achieves lower average MSE than both benchmarks for a given experiment and AL iteration ($p < 10^{-33}$, Appendix C). Table 1 illustrates the substantial data efficiency gains produced by DEIMOS: both alternative methods require over $65\%$ more data than DEIMOS to achieve a test MSE of 0.10, and both require at least $75\%$ more data than DEIMOS to achieve a test MSE of 0.095.

Maximum variance acquisition performs poorly throughout, indicating that acquiring uncertain points does not necessarily lead to effective AL on its own. The strong performance of DEIMOS compared to maximum variance acquisition demonstrates the benefit over traditional uncertainty sampling approaches gained by modeling correlations between predictions across data points and assessing EI throughout the input space.

**Face age prediction.** AL experiments were run on the UTKFace dataset (Fig 2 center and 2 right), with models trained to predict the age corresponding to each $200 \times 200$ pixel face. DEIMOS performed better than random and maximum variance acquisition in these experiments, with its root MSE (RMSE) being significantly lower than that of the other acquisition functions for a given experiment and AL iteration. Notably, DEIMOS outperforms both benchmarks even for moderately large batch sizes (100, 200), providing empirical evidence for the effectiveness of the proposed batch-mode algorithm in querying informative, non-redundant points.

### 5.3 ACTIVE LEARNING EXPERIMENTS: CNN CLASSIFICATION

We evaluated DEIMOS on MNIST (LeCun et al., 1998) for binary classification of 7s vs 9s and multiclass classification of all 10 digits. We compare to random acquisition, acquisition of the point(s) with maximum entropy (Gal et al., 2017), BALD/Batchbald (Houlsby et al., 2011; Kirsch et al., 2019), and core-set AL via robust k-Center (Sener & Savarese, 2018).

In binary classification of 7s and 9s, DEIMOS compares favorably to existing approaches. For single-point acquisitions, DEIMOS requires roughly one-fourth as much data as random acquisition to arrive at test accuracy thresholds of 0.97 and 0.975, and shows data efficiency improvements over all other benchmarked methods (Table 2, Appendix B). For batch size 25 (Fig 3(a)), DEIMOS performs substantially better than random acquisition throughout the experiment and improves upon Batchbald and robust k-Center. The strong performance of DEIMOS is robust to the choice of representative sample size: there is no statistically significant difference in DEIMOS AL performance

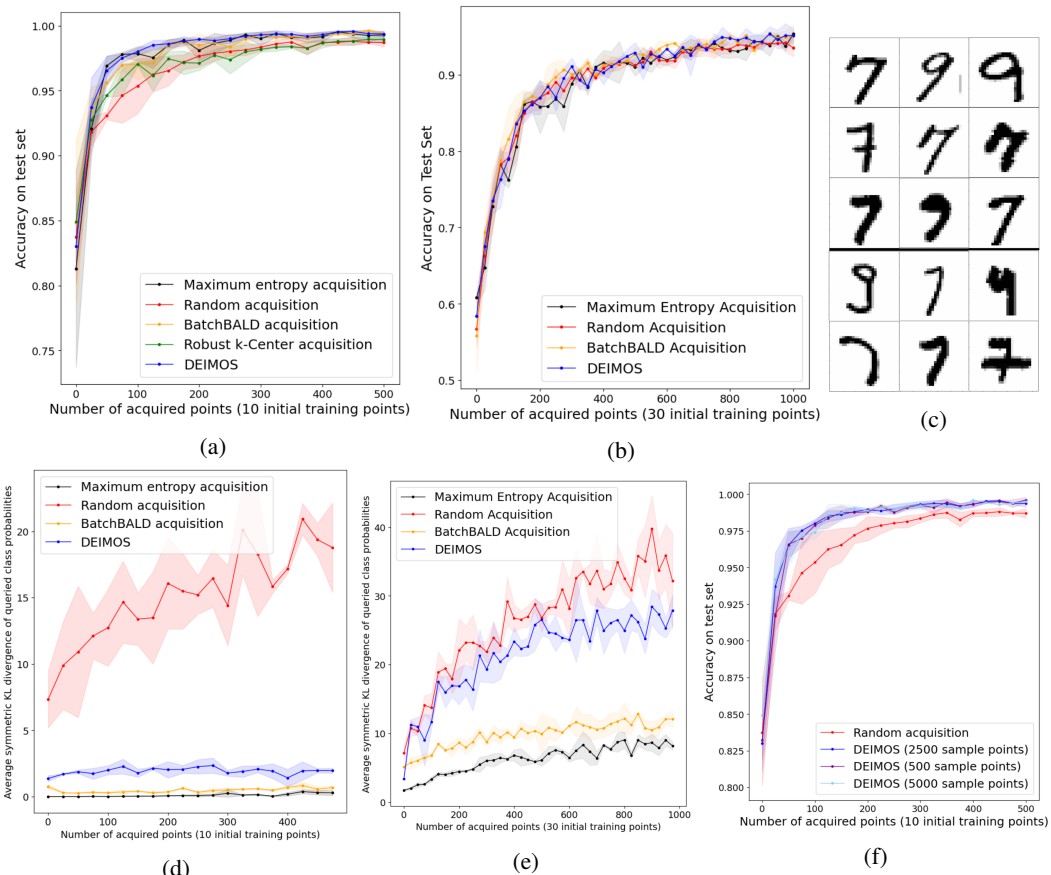

Figure 3: **(a)** MNIST 7 vs. 9 classification accuracy ($\pm 1$ standard deviation shaded) across AL iterations for batch size 25. DEIMOS outperforms random and robust k-Center acquisition ($p$-values $4.4 \times 10^{-11}$ and $1.9 \times 10^{-11}$, respectively (Appendix C) **(b)** MNIST 0-9 classification accuracy ($\pm 1$ standard deviation shaded) across AL iterations for batch size 25. DEIMOS outperforms random ($p = 9 \times 10^{-3}$) and maximum entropy ($p = 3 \times 10^{-3}$) acquisition (Appendix C) **(c)** Images 1-3 and 999-1000 (order: top to bottom) queried by DEIMOS in MNIST 7 vs. 9 classification with batch size 25; each column corresponds to one experiment (order: left to right). **(d,e)** Batch diversity, defined as average symmetric KL divergence of predicted class probabilities for queried points in each batch, across AL iterations in MNIST 7 vs. 9 classification with batch size 25 and MNIST 0-9 classification with batch size 25, respectively. DEIMOS has higher batch diversity than Batchbald and maximum entropy acquisition ($p < 0.01$) in almost all AL iterations in both tasks **(f)** DEIMOS MNIST 7 vs. 9 classification accuracy across AL iterations (batch size 25) for several representative sample sizes $|X_{\text{samp}}|$. There is no statistically significant difference in DEIMOS AL performance across the $|X_{\text{samp}}|$ shown (Appendix C).

Table 2: Top: Number of single-point acquisitions required in order to obtain the MNIST 7 vs. 9 classification test accuracy listed in the left column for each acquisition function (Appendix B). Bottom: $p$-values comparing benchmark accuracy to DEIMOS accuracy (Appendix C).

| Test accuracy | DEIMOS | BALD | Max. entropy | Random | Robust k-Center |
|---|---|---|---|---|---|
| 0.95 | 20 | 23 | 21 | 48 | 24 |
| 0.96 | 24 | 25 | 31 | 55 | 39 |
| 0.97 | 27 | 39 | 39 | 113 | 58 |
| 0.975 | 41 | 67 | 58 | 158 | 58 |
| $p$-value | - | $2.3 \times 10^{-21}$ | $4.1 \times 10^{-11}$ | $5.2 \times 10^{-133}$ | $7.7 \times 10^{-46}$ |

for representative set sizes of 500, 2500, and 5000 (Fig 3(f)). Fig 3(c) illustrates the first three and last two images queried by DEIMOS in the batch size 25 experiments. Early on, DEIMOS queries class prototypes (e.g. first image queried in experiment 1) and probes the decision boundary (e.g. first and third images queried in experiment 2). When the training set is larger, DEIMOS focuses more on querying outliers (e.g. last two images requested in experiment 1) that may only clarify a narrow subset of predictions.

In MNIST classification of all 10 digits (Fig 3(b)), DEIMOS is comparable to state-of-the-art methods. For batch size 25, DEIMOS outperforms random and maximum entropy acquisition ($p < 9 \times 10^{-3}$). Batchbald slightly outperforms other methods early on in the experiments but DEIMOS is comparable to Batchbald in most AL iterations. The strong performance of DEIMOS demonstrates the advantages of maximizing EI for all classes over a large sample of points and taking into account correlations between predicted class probabilities across points.

The diversity of batches queried by DEIMOS was analyzed in the MNIST 7 vs. 9 and 0-9 classification experiments for batch size 25 (Fig 3(d), 3(e)). Batch diversity is measured by the average symmetric KL divergence in the predicted class probability distributions for pairs of points in the queried batch. Across both tasks, batches of points queried by DEIMOS had higher batch diversity ($p < 0.01$) than those queried by Batchbald and maximum entropy acquisition in virtually all AL iterations. Random acquisition led to batches with the highest diversity, in large part due to the relatively high prevalence of pairs of points with very low predictive uncertainty that are predicted by the model to belong to different classes (leading to high symmetric KL divergence) in randomly-assembled batches. Such points are less likely to be queried by other AL methods as they are already predicted confidently and therefore are often uninformative. Figures 3(d) and 3(e) show the effectiveness of the DEIMOS batch-mode AL algorithm in assembling heterogeneous batches of points.

## 6 CONCLUSION

We formulate an active learning approach, DEIMOS, aimed at maximizing expected improvement by minimizing expected predictive variance. Our approach extends to the batch-mode setting when supplemented with an efficient greedy algorithm that sequentially assembles a diverse batch of queried points. The effectiveness of the proposed algorithm is studied across multiple classification and regression tasks, and it is found to outperform typical baselines in regression and to achieve performance on par with–and in some cases better than–existing methods in classification.

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

## A    OPTIMALITY OF DEIMOS BATCH-MODE ACQUISITION

The proposed greedy algorithm for batch-mode active learning is an efficient alternative to evaluating the EI for all possible batches of unlabeled candidate points of specified size and selecting the batch that maximizes EI. Although the proposed algorithm is efficient, its greedy approach means that it does not always find the batch of points that maximizes EI. An example of $\text{Var}_q(\hat{Y}_{\text{samp}}(\theta))$ where greedy batch-mode active learning may not maximize EI is provided below:

$$\text{Var}_q(\hat{Y}_{\text{samp}}(\theta)) = \begin{bmatrix} 9 & 3 & 2 \\ 3 & 2 & 3 \\ 2 & 3 & 9 \end{bmatrix}$$

For the given $\text{Var}_q(\hat{Y}_{\text{samp}}(\theta))$, DEIMOS batch-mode acquisition with batch size 2 queries point 2 and then either point 1 or point 3 (both options have equal expected improvement), resulting in a total variance reduction of approximately 16.9. However, the optimal trace reduction (i.e. expected improvement) of roughly 19.6 results from querying points 1 and 3. Therefore, in this example DEIMOS achieves only around 86% of the optimal trace reduction for matrix $\text{Var}_q(\hat{Y}_{\text{samp}}(\theta))$.

Next we explore how sub-optimal the greedily built batches tend to be for randomly generated variance matrices. The empirical effectiveness of the proposed approach for sequential batch assembly is compared to that of searching through all batches of points and acquiring the batch that maximally reduces predictive variance. We simulate i.i.d. standard normal predictions for a specified number of pool points and dropout masks. $\text{Var}_q(\hat{Y}_{\text{samp}}(\theta))$ is calculated using the sample covariance between all points, $\text{Var}_q(\hat{Y}_{\text{samp,dropout}}(\theta))$, and adding $\tau^{-1}I$ (equation 3), where $\tau^{-1}$ is set to 10% of the average MC dropout sample variance across all points. Synthetic prediction variance matrices generated in this fashion are used to compare the expected improvement from the greedy algorithm, given by the trace reduction in the matrix, to that resulting from acquiring the optimal batch of points.

The procedure detailed above is repeated to generate 200 instances of $\text{Var}_q(\hat{Y}_{\text{samp}}(\theta))$ across batch sizes of 2, 5, and 10 and for varying numbers of i.i.d. normal variables designated as the number of model predictions per point (i.e. the number of dropout masks). For each $\text{Var}_q(\hat{Y}_{\text{samp}}(\theta))$, the DEIMOS batch-mode algorithm was run and the ratio of the resulting variance trace reduction to the optimal variance trace reduction, computed by searching through all possible batches of points, was computed. Although the scope of such experiments is limited by the exponential time complexity of iterating through all possible batches of points, the trace reduction ratios are typically quite close to 1 throughout our experiments (Figure A.1). Thus, while instances of $\text{Var}_q(\hat{Y}_{\text{samp}}(\theta))$ where the greedy algorithm deviates significantly from the optimal trace reduction do exist, they appear to be relatively uncommon. These results provide some empirical support for the effectiveness of the proposed greedy algorithm in achieving a variance reduction across predictions comparable to that resulting from querying the optimal batch of points under our assumptions.

## B    MNIST BINARY 7 VS. 9 BATCH SIZE 1 RESULTS

We show that DEIMOS outperforms several existing active learning methods in MNIST binary classification of 7 vs. 9. With single-point acquisitions and initial training sets consisting of 10 points, DEIMOS performs substantially better than random acquisition, and noticeably better than robust k-Center and BALD, throughout most of the active learning iterations (Figure B.1). Maximum entropy acquisition also performs well, achieving performance comparable to that of DEIMOS. Overall, DEIMOS performs well compared to existing active learning methods for classification in the MNIST 7 vs. 9 classification task.

## C    HYPOTHESIS TESTING FOR AL PERFORMANCE EVALUATION

We used hypothesis tests based on linear mixed models (LMMs) to compare the performance of different AL algorithms in our experiments. If two AL algorithms are equally effective, then predictive performance on test data will be given as a function of the number of acquisitions (the algorithm in use has no effect). However, if one active learning algorithm is more effective than another, predic-

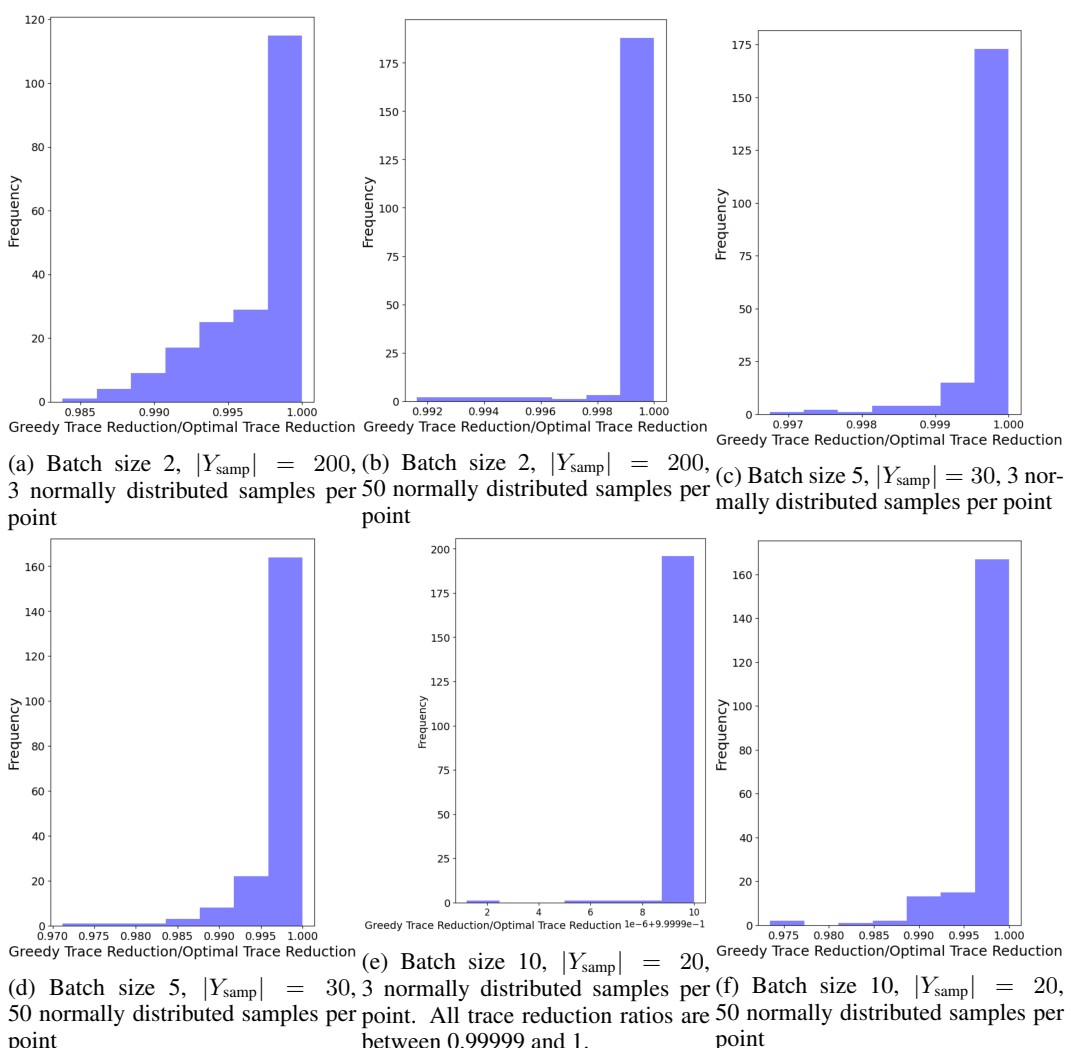

Figure A.1: Histograms of the ratio of the variance trace reduction resulting from applying the proposed greedy algorithm to the variance trace reduction resulting from querying the optimal batch of points, under our assumptions. Each trial uses a different $\text{Var}_q(\hat{Y}_{\text{samp}}(\theta))$ initialized by calculating $\text{Var}_q(\hat{Y}_{\text{samp,dropout}}(\theta)) + \tau^{-1}I$ from i.i.d. normally-distributed variables, where a specified number of normal samples corresponds to the model's MC dropout predictions for each point. All ratios of greedy trace reduction to optimal trace reduction are above 0.97 across all experiments.

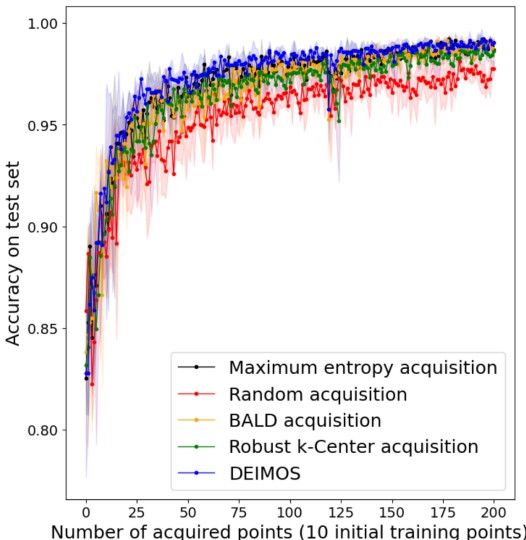

Figure B.1: MNIST 7 vs. 9 classification test accuracy ($\pm$ 1 standard deviation shaded) averaged over three active learning experiments as a function of number of acquired points, with batch size 1.

tive performance on test data will have contributions both from the algorithm in use and from the number of acquisitions that have occurred.

Consequently, the null model for predictive performance during an AL experiment is given by an LMM with random-effect terms corresponding to the number of elapsed acquisition iterations. The alternative model for predictive performance is an LMM with a fixed-effect term corresponding to the AL algorithm and random-effect terms corresponding to the iteration.

A likelihood ratio test is used throughout our experiments to compare the null model and alternative model and determine if the fixed effect corresponding to the choice of AL algorithm is statistically significant.

## D  ACTIVE LEARNING EXPERIMENT DETAILS

**Synthetic 1D neural network regression.**   1D real-valued output is generated from input using a random neural network with zero-mean Gaussian weights and biases and fully-connected hidden layers of size $[32, 32]$; observed output is corrupted by zero-mean, i.i.d. Gaussian noise. A fully-connected neural network with hidden layers of size $[256, 256, 256]$, with $p = 0.2$ and $\lambda = 0.0005$, is fit to the synthetic data and used to evaluate EI.

**Active learning experiments.**   Active learning experiments are run on individual Tesla P100 and Tesla V100 GPUs. Across all AL experiments, acquisition functions are calculated using 50 MC dropout predictions for each point. In the interest of computational efficiency, only a specified number of unlabeled points are considered for acquisition in each iteration (as opposed to considering all of $X_{pool}$). In DEIMOS acquisition, the set of candidate points for acquisition in each AL iteration is restricted to the unlabeled points in $X_{\text{samp}}$. $X_{\text{samp}}$ is randomly sampled from $X_{\text{train}} \cup X_{\text{pool}}$ until the set of candidate points for acquisition $X_{\text{cand}} = X_{\text{samp}} \setminus X_{\text{train}}$ reaches a specified size.

**Alternative splicing.**   Convolution is performed on the sequence input by one-hot encoding the 101-nucleotide sequence as a 2-dimensional array with four channels corresponding to the four possible nucleotides at each position. The dataset contains a total of 265,137 examples, of which 10% are used as the test data and the remaining 90% is subdivided into the training data and the pool (from which a validation set consisting of 50 points is sampled). In the AL experiments (Fig 2 left), all algorithms begin with training sets of size 75 and acquire 600 points from the pool through single-point acquisitions. The number of unlabeled pool points considered for acquisition in each AL iteration is 5,000 for all algorithms tested.

**UTKFace face age prediction.**  Initial training sets consisted of 150 images, and 2,000 additional images were acquired over the course of the experiments. All 23,708 images in the dataset were used, with $10\%$ held out as test data and the remaining $90\%$ consisting of the training data and the pool (from which a 50-image validation set is taken). During each acquisition iteration points are acquired from a set of 5000 candidates randomly chosen from the pool set (for all algorithms tested).

**MNIST handwritten digit classification.**  In the AL experiments, 10,000 images are used as test data while 60,000 images are used for the training data and the pool (from which a validation set of 10 images in binary classification/30 images in 0-9 classification is sampled). In each AL iteration, points are queried from a randomly selected set of pool candidates; 2500 and 2000 candidates are considered per iteration in 7 vs. 9 and 0-9 classification, respectively. All algorithms begin with an initial training set of 10 images in binary classification experiments and 30 images in 0-9 classification experiments. In our experiments, initial training sets have equal representation from all classes.

## E  HYPERPARAMETER TUNING

Across all experiments, the dropout probability $p$ is calibrated such that the resulting model makes relatively high quality predictions and MC dropout uncertainty estimates on the validation set. Quality of MC dropout uncertainty estimates is assessed by Pearson $R^2$ between the observed squared error vs. predicted variance on the validation set.

L2 regularization is applied in fully-connected layers with weight decay declining throughout the active learning experiments according to $\lambda = C/N$, where $C$ is a weight decay hyperparameter tuned on the validation set and $N$ is the size of the training set. $C$ is selected based on validation set accuracy resulting from model training on the initial training set for a given number of epochs with the corresponding $\lambda$. With this setup, the model precision in regression $\tau$ is constant across active learning iterations ($\tau_s$ is a constant in classification).

In regression experiments, the model precision hyperparameter $\tau$ is set such that $\tau^{-1}$ is between $0.1\bar{\sigma}^2_{reg,v}$ and $0.2\bar{\sigma}^2_{reg,v}$, where $\bar{\sigma}^2_{reg,v}$ is the average dropout variance predicted by the initial model (excluding $\tau^{-1}$) on the validation set.

In classification experiments, $\tau_s$ is set such that $\tau_s^{-1}$ is between $0.001\bar{\sigma}^2_{class,v}$ and $0.01\bar{\sigma}^2_{class,v}$, where $\bar{\sigma}^2_{class,v}$ is the average dropout variance across classes predicted by the initial model on the validation set. The scaling factor multiplying $\bar{\sigma}^2_{class,v}$ to determine $\tau_s^{-1}$ should be much less than 1 given that $\tau_s^{-1}$ is merely a smoothing parameter aimed at ensuring stable matrix inversion of $\mathrm{Var}_q(\hat{p}_{new}(\theta))$.

All hyperparameter values used to generate results can be found in the corresponding code[2].

---

[2]Code for all experiments is included in the supplementary material.

