# OpenReview forum: "Active Learning in CNNs via Expected Improvement Maximization"
_ICLR.cc/2021/Conference — Reject_

### Official Review · AnonReviewer1 · 2020-10-26
**Official Blind Review #1**

**Rating:** 4
**Confidence:** 4

**Review:**

The paper considers the active leaning problem by proposing a new acquisition function based on Expected improvement (EI). The paper shows that acquiring the point maximizing the expected reduction in predictive uncertainty across all points is equivalent to maximizing the expected improvement (EI).


The paper considers the setting for CNN classification and regression where the authors derive the variance estimations. The paper also extends to the batch mode to select multiple points simultaneously. The uncertainty is estimated using MC dropout.

The papers show that the improvement in the experiment is not significant comparing to the baselines.

=============================
Major concern: although the paper claims to derive the EI-based acquisition function for active learning. The resulting derivation shows that the final acquisition function only depends on the uncertainty of the training data given the select point (x_new) while the relative contribution of the x_new toward the final performance E(y_train | x_new) has been vanished. Note that in the original form of the EI (Jones et al 1998), the acquisition function does not only depend on the uncertainty, but also the expected function value at the x_new.

The reviewer thinks that it may be better for this paper to position the contribution (abstract/introduction…) as the uncertainty based approach for active learning, then claim the minor/secondary contribution as showing the connection to the EI for the active learning setting.

At the current form, the reviewer thinks the paper is under the acceptance threshold.


Minor concern:

The paper claims that they utilize full joint covariance rather than just point-wise variances used in previous work. However, the reviewer thinks this may be a wrong claim. In particular, the trace of variance considered in Eq 2 is equivalent to point-wise variance.



Other comments:

The intuition in the resulting acquisition function is very much related to the Predictive Variance Reduction Search [1] – it is worth mentioning. Both approaches rely on:
(1) uncertainty estimation of the some targets;
(2) the objective function is defined using the sum of the uncertainty reduction – the trace of variance in Eq (2) and the original form in Eq (1) share this spirit.
(3) similar in the batch setting where both approaches will sequentially select point to fill in a batch.

The difference is that [1] considers another problem in Bayes Opt while this paper considers the active learning task.



[1]  Nguyen, V., Gupta, S., Rana, S., Thai, M., Li, C., & Venkatesh, S.  Efficient Bayesian Optimization for Uncertainty Reduction Over Perceived Optima Locations. In IEEE International Conference on Data Mining (ICDM) (pp. 1270-1275). 2019.

---

> ### Author Response · Authors · 2020-11-25
> **Authors' Response to Reviewer 1**
>
> We thank the reviewer for their comments.
>
> Regarding our contribution: Our main contribution is not deriving the EI; as the reviewer observes, EI-based algorithms have been formulated for Bayesian optimization. We view our main contribution as extending integrated variance approaches, which attempt to construct statistically optimal active learning queries, to CNNs.
>
> In response to the major concern: We believe that the reviewer may have misunderstood our definition of expected improvement (see Eq 5 and Eq 6 and the paragraph following each of these two equations). The notion of expected improvement used in our paper is different from that used in (Jones et al 1998) and, consequently $E(y_{train} | x_{new})$ does not appear in our objective function. As stated in the paper, we define expected improvement as the reduction in predictive variance across a large, representative sample of points that is expected to result from querying x_{new} under our assumptions. Section 3.1 of the paper shows that the $x_{new}$ maximizing expected improvement (as we define it) is the same $x_{new}$ that minimizes expected prediction error after it is queried. Since our definition of the expected improvement involves the variance in the prediction of Y rather than the value of Y itself, the expected value of Y does not come into play.
>
> In response to concerns that the improvement is not significant compared to the baselines: Multiple experiments presented in the paper show that the proposed algorithm outperforms the baselines by a statistically significant margin. All p-values comparing performance of our active learning method to the baselines in the regression setting are below 0.01. The proposed algorithm has outperformed random acquisition and maximum variance acquisition by a statistically significant margin in all active learning experiments we have conducted for regression tasks. Additionally Table 2 shows that the proposed approach performs better than all baselines, including state-of-the-art methods such as core-set active learning, in MNIST 7 vs. 9 classification with batch size 1. Although the proposed approach does not exhibit statistically significant improvement over all baselines in MNIST 0-9 classification for batch size 100, it is on par with top-performing existing methods. In MNIST 0-9 classification with batch size 25, the proposed method outperforms random and maximum entropy acquisition ($p<0.01$, Figure 3(b)).
>
> In response to the minor concern: We believe the reviewer may have misunderstood the proposed active learning approach. Our method does, importantly, utilize the full joint covariance of model predictions across the representative sample of points. It is correct that Eq 2 states that the point acquired minimizes the point-wise variance such that it minimizes expected squared prediction error. However, the way in which we choose $x^* $ and estimate the right hand side of Eq 2 involves the covariance between model predictions, as is discussed in Sections 3.3 and 3.4. We believe that acquiring points based on covariance between model predictions is an important advantage of our method. For instance, uncertainty sampling methods typically query uncertain points, but no claim can be made about the resulting improvement in model predictions. On the other hand, our formulation utilizes the joint covariance of model predictions across a large sample of points to query the point that, accounting for correlations between model predictions across points, minimizes expected prediction error upon being queried.
>
> Thank you for pointing us to the paper on Predictive Variance Reduction Search; it is certainly relevant and we will further revise the paper to discuss it.

---

### Official Review · AnonReviewer3 · 2020-10-27

**Rating:** 6
**Confidence:** 3

**Review:**

Paper Summary

The paper considers the problem of active learning for training convolutional neural networks (CNN) in a sample-efficient manner. The proposed approach is built upon the existing idea of selecting points that maximally reduce expected mean squared error (MSE) on a large representative sample of points. MC-dropout is used for obtaining the estimates of model uncertainty. This idea is used for active learning in regression and classification problems with CNNs. A greedy method is proposed to select a batch of points by maximizing the acquisition function score sequentially obtained by updating the covariance matrix on previous points selected in the batch. Experiments are performed on two regression and one classification task.


Detailed Comments

- The paper tackles an important problem relevant for many practical applications. Although the proposed approach is based on an existing idea, the application to CNNs specifically seems novel.

- It is a little worrying to see that random acquisition performs equally well with the proposed acquisition function (Figure 2 for instance). Please provide more discussion on this point and/or run one experiment for more than 3 (current) random runs to get a clear picture. Random acquisition is a simple to implement method and if the gains by the proposed acquisition function are not large enough, random acquisition will be a preferable option for a user.

- There is limited description about how is the ''representative sample chosen'' in the experimental section. Please provide more details on this.

- Greedy batch selection methods are known to select similar points losing the batch advantage. Please provide more details on the diversity of the points in batch selection.

- Why MC-dropout specifically is used for uncertainty estimation? The proposed approach should ideally be agnostic to the method for uncertainty estimation. Please provide more discussion on this choice.

- In my opinion, although it is a minor point, it is better to show the visualization of EI acquisition function on a standard synthetic function like a sinusoidal function, Branin function etc. instead of a random neural network which might seem a little contrived.

- The writing of the paper can be improved. For example, the usage of the term 'Expected improvement' is spread out across the paper without a proper technical description. Please try to provide a clear technical description of all the key concepts

---

> ### Author Response · Authors · 2020-11-25
> **Authors' Response to Reviewer 3**
>
> We thank the reviewer for their thoughtful feedback.
>
> Thank you for the feedback regarding experiments in which the proposed algorithm does not significantly outperform random acquisition. Figure 2, which was mentioned in your feedback, in fact shows that in the regression setting the proposed method does consistently outperform random acquisition by a statistically significant margin. However, MNIST 0-9 classification with batch size 100 is a task where many active learning methods, including the proposed approach, do not outperform random acquisition significantly. We have replaced the MNIST 0-9 classification results for batch size 100 with results from MNIST 0-9 classification with batch size 25 (Figure 3(b)), in which our method outperforms maximum entropy acquisition and random acquisition ($p<0.01$). We will revise Figure 3(b) to include the performance of robust k-Center acquisition.
>
> The procedure by which the representative sample of points is chosen in our experiments is detailed in Appendix D. The representative sample of points is simply a random sample of specified size from $X_{train} \cup X_{pool}$. We will incorporate these details into the main paper to make the discussion of our experiments clearer.
>
> It is true that greedy batch selection methods often query correlated points, which results in reduced performance. Although our batch-mode active learning algorithm is a greedy algorithm, querying batches of diverse points, via iterative updates to the predictive covariance, is one of its key advantages. The proposed batch-mode active learning algorithm sequentially assembles a batch by adding the point that is expected to maximally reduce predictive variance conditioned on all existing points in the batch. Therefore, we explicitly construct batches such that each new point added to a batch is expected to maximally reduce the predictive uncertainty that remains after querying the existing points in the batch. We have found empirically that the points queried by the proposed method are diverse even in batch-mode active learning with moderately large batch sizes (see, for instance, Figure 3(c)). In order to further analyze the proposed method’s effectiveness in assembling diverse batches of points, we have run new experiments on MNIST 7 vs. 9 and 0-9 classification (batch size 25) analyzing batch diversity as measured by the average symmetric KL divergence between the predicted class probability distributions of pairs of images in the queried batch. In both classification tasks, we found that EI acquisition has a far higher batch diversity than alternative methods including Batchbald and maximum entropy acquisition ($p<0.01$) at virtually every stage in the active learning experiments. The paper has been updated to include discussion of the results from these batch diversity experiments (Figure 3(d), 3(e); Section 5.3). We will further revise the paper to include core-set AL in our batch diversity analysis.
>
> MC dropout is used for uncertainty estimation because (i) it is computationally efficient compared to ensemble methods (ii) it enables estimation of covariance between different model predictions as opposed to simply estimating the point-wise variance and (iii) can be used even for pretrained models that used dropout.
>
> Thank you for the feedback regarding the writing. We will revise the paper such that the definition of expected improvement is made explicit early on.

---

### Official Review · AnonReviewer4 · 2020-10-28
**The paper combines dropout-based variational inference and integrated variance minimization**

**Rating:** 6
**Confidence:** 4

**Review:**

The paper proposes a method for pool-based active learning in CNNs, selecting the next (batch of) data from an unlabeled pool to query their labels to expedite the learning process. The method computes the expected reduction in the predictive variance across a representative set of points and selects the next data point to be queried from the same set. In batch settings, the data points are sequentially selected in a batch (in a greedy way), with predictive variance representation updated after each selection. Experiments are performed on MNIST classification, and regression tasks of alternative splicing prediction, and face age prediction.

Pros:

The method is rather simple and (up to some extent) computationally efficient. The paper is generally well-written and puts the proposed method into context. The proposed method performs better than maximum variance acquisition and random acquisition in the regression experiments. For the classification tasks, its performance is comparable with maximum entropy, batch BALD and robust k-Center acquisition.

Concerns:

Overall, the novelty of the paper is limited. It employs dropout-based variational inference and applies the integrated variance minimization idea. The main contribution seems to be considering a joint normal distribution (whose level of validity is not completely clear/discussed). It is not clear if the proposed acquisition is the statistically optimal one that minimizes the expected MSE. It seems to me that requires taking into account the distribution of the (unobserved) label. Also, assuming all predicted class probabilities across all sample points as jointly Gaussian is strange as the probabilities are bounded and should sum to 1. In fact, the performance comparison in the classification setup is limited to one dataset and the results do not seem to confirm that the proposed method outperforms the baselines.

Figure 2 seems to suggest that with increasing batch size, the performance gap shrinks. Have the authors tried a larger batch size or checked the sensitivity to it?

Additionally, how sensitive is the performance to the number of randomly sampled data points for the representative set?

My current rating is based on the aforementioned concerns.

Update:
I thank the authors for their response. I read the authors’ responses and the updated paper. As the authors have addressed some of my concerns and questions, I have adjusted my rating. However, I still have my concerns regarding the novelty and methodological contribution, and the classification setup. I also agree with other reviewers that the presentation can be improved.

---

> ### Author Response · Authors · 2020-11-25
> **Authors' Response to Reviewer 4**
>
> We thank the reviewer for their constructive feedback.
>
> The main contribution of our work is extending integrated variance-based approaches to active learning to CNNs. Previously, integrated variance-based approaches have typically been applied to Gaussian processes (GP) rather than neural networks. The joint Gaussian assumption can be motivated as approximating the CNN by a GP thereby enabling tractable estimation of the variance of a large, representative sample of points conditioned on an unlabeled data point.
>
> Under the assumptions stated in the paper, the proposed acquisition function is the statistically optimal one, in that the queried point minimizes expected MSE conditioned on that point. This is shown in Section 3.1. The distribution of the unobserved label does not affect the validity of the argument presented in Section 3.1 because the distribution of the unobserved label only affects the second and third terms of the expansion in Eq (1), both of which are constant in $x_{new}$.
>
> The proposed approach is motivated by the regression setting, for which relatively few active learning algorithms have been proposed for neural networks. The presentation of the classification setting along with empirical results is intended to demonstrate that the method extends to classification, although that is not our main contribution. You are correct in noting that the joint Gaussian assumption in the classification setting is an approximation given that probabilities are bounded. As stated in the paper, we investigated alternative formulations of the proposed approach in classification tasks such that the joint Gaussian assumption is applied to unbounded quantities such as logits. However, our empirical results demonstrated that our method in classification was most effective when applied to probabilities in spite of the apparent limitations of this approximation. We address the approximate nature of the joint Gaussian assumption in classification in the third paragraph of Section 3.4 of the paper.
>
> We have observed that our method performs well compared to the baselines for batch sizes of up to several hundred points. The method tends to outperform baselines by a lesser extent for batch sizes greater than several hundred points. However, we believe that the strong performance of the proposed method for batch sizes up to several hundred points is sufficient for it to be useful in a variety of applications.
>
> We have found empirically that performance is not very sensitive to the number of points in the representative set. We have run new experiments on MNIST 7 vs. 9 binary classification in order to gauge the sensitivity of the proposed method to the representative sample size. Hypothesis tests based on linear mixed models for batch size 25 found no statistically significant difference in the active learning performance of our method run with representative sample sizes of 500, 2500, and 5000 points (and for all representative sample sizes, our method significantly outperformed random and robust k-Center acquisition). These results have been added to the paper (Figure 3(f)).

---

### Decision · Program_Chairs · 2021-01-07
**Final Decision**

**Decision:**

Reject

**Comment:**

This paper proposes an approach for active learning in CNNs. The method computes the expected reduction in the predictive variance across a representative set of points and selects the next data point to be queried from the same set.

Pros:
- The method is rather simple and seems practical.
- The paper is generally well-written.

Cons:
- The novelty of the paper is limited, as it essentially applies a known approach to CNNs.
- The performance gains presented in experiments seem rather mild, and may not justify using this method.